# Development of a Wet-Granulated Sourdough Multiple Starter for Direct Use

**DOI:** 10.3390/foods11091278

**Published:** 2022-04-28

**Authors:** Giuseppe Blaiotta, Raffaele Romano, Marco Trifuoggi, Maria Aponte, Agnese Miro

**Affiliations:** 1Department of Agricultural Sciences, University of Naples Federico II, Via Università 100, 80055 Portici, Italy; blaiotta@unina.it (G.B.); raffaele.romano@unina.it (R.R.); 2Department of Chemical Sciences, University of Napoli Federico II, Via Cinthia, 80126 Naples, Italy; marco.trifuoggi@unina.it; 3Department of Pharmacy, University of Napoli Federico II, Via Domenico Montesano 49, 80131 Naples, Italy; miro@unina.it

**Keywords:** sourdough, starter, wet granulation, volatile compounds, GC/MS

## Abstract

The search for sourdough starters for the direct production of baked goods with all the advantages of biological sourdough fermentation is still a crucial issue. In this study, 43 Lactic Acid Bacteria strains isolated from mature sourdoughs were evaluated for features of technological interest and tested for fermentation ability. Three microbial combinations were selected and used to produce bread. Based on GC-MS and sensory analysis, bread made by using the three combinations of strains was characterized by a more complex aroma profile with the prevalence of VOCs typical of sourdough bread. To set up the best way to keep microbial viability upon drying, the three combinations were subject to freeze-drying and wet granulation, with the latter being used for the first time for food starters’ stabilization. Wet granulation ensured optimal strains’ viability. Surprisingly, the height attained by mature sourdoughs when inoculated with wet granulated starters was constantly higher than the height reached by sourdoughs made with the same starters as fresh cells. The microbial combination E75-B72 exhibited the best performances and may represent a starter able to ensure sourdough bread production in 16 h of fermentation at 28 °C.

## 1. Introduction

The use of the sourdough process as a means of leavening is one of the oldest biotechnological processes in cereal food production. Sourdough is a mixture of mainly cereal flour and water, made metabolically active by a heterogeneous population of Lactic Acid Bacteria (LAB) and yeasts, either by spontaneous fermentation or by fermentation triggered by the addition of a sourdough starter culture, whether or not involving back-slopping [1]. Traditional bread making with ‘natural yeast’ takes time and it is quite difficult to keep it stable over time. In fact, the use of sourdough gradually fell into disuse and was replaced by industrial yeast. Many scientific papers have highlighted the valuable properties of sourdough-based bakery products, sparking renewed interest in this type of treatment around the world. Most of the rheological, sensory, and nutritional features, as well as the extended shelf-life, are to be related to microbial metabolic activities, such as fermentation, proteolysis, synthesis of volatile compounds, and antifungal activity [2]. 

The maintenance of traditional type I sourdoughs is time-consuming and requires hardworking and skilled labor. The sourdough ecosystem can be roughly mimicked by combining properly selected starter cultures. The so-called type II sourdough is produced by a process known as the industrial method as it involves a single fermentation step of LAB only or LAB with yeast, over 15–24 h, and then back-slopped ones [3]. As a result of the high initial load, the starter culture may dominate or inhibit the growth of the autochthonous dough microbiota. Type III sourdough is produced by dehydrating the stabilized form of type II sourdough [3]. Despite several advantages that make type II sourdough ideal for use in industrial processes, its application is jeopardized by the low survival rate of the strains during storage, which is usually achieved by drying [4]. Companies marketing type III sourdough often do not ensure the viability of the sourdough microbiota, so this kind of product is more useful to improve the texture and aroma of the final products, but the addition of baker’s yeast to allow the leavening is required in any case [4]. 

The development of sourdough powder holds several advantages over fresh sourdough including a longer shelf-life, consistent product quality, ease of formulation and mixing, and lower transportation costs [5]. Drying techniques such as spray-drying and freeze-drying have been used to achieve a longer shelf-life of sourdough and turn it into a stock product until further use [5]. Dried sourdough cannot be used directly for dough formulation due to injured, stressed, and reduced number of cells [6]. Pretreatments such as powders’ rehydration and LAB refreshment have been shown to be necessary to activate dried sourdough starter [5]. Wet granulation, a well-known technique in the pharmaceutical industry, is a key unit process for the manufacturing of solid dosage forms, especially tablets and capsules [7]. In food technologies, wet granulation has been used to produce granola, an aggregated baked food product commonly eaten as a breakfast cereal [8]. Apart from granola, no further efforts have been made for evaluating the behavior of food ingredients treated with this technique and, as matter of fact, food applications of wet granulation are still at an infant stage. On the other hand, wet granulation has been successfully used for the stabilization of the probiotic *Lactiplantibacillus plantarum* subsp. *plantarum* strain 299v, by employing as granulating/binding agents, excipients generally regarded as safe [9]. Moreover, wet granulation was used for the stabilization of *Limosilactobacillus reuteri* LR6 cells. In this case, the minimal reduction in strain viability was obtained for cells granulated in corn starch or microcrystalline cellulose [10]. 

Starter selection has become important to improve the technological characteristics of bakery products to meet the growing industry and consumer demand for developing new products using sourdough. Such selection is conventionally based on the cultures’ capacity to rapidly acidify the flour-water mixture and/or on the ability to produce specific flavors [4]. Actually, the search for sourdough starter cultures for the direct production of baked goods of consistent quality and with all the advantages of biological sourdough fermentation, such as aroma and taste, freshness retention, and extended microbial shelf-life, is still a crucial issue. In this study, 43 LAB strains isolated from mature sourdoughs obtained by different flour types during a previous survey [11] were evaluated for features of technological interest and tested for the fermentation ability in model systems. The best combinations were subject to several protocols in order to point out the best way to keep microbial viability upon drying. Specifically, wet granulation was used for the very first time for starter cultures stabilization. Outcomes related to starter’s viability upon drying and to leavening performances were compared with those obtained by freeze-drying, the technique usually adopted for stabilizing sourdough starters [5]. 

## 2. Materials and Methods

### 2.1. Microorganisms and Culture Conditions

A total of 43 LAB cultures were isolated by mature sourdoughs during a previous survey. Strains were identified and genotypically characterized [11,12]. 

All cultures were stored at −25 °C in liquid MRS (Oxoid, Basingstoke, Hampshire, UK) with 20% (*v*/*v*) sterile glycerol (Merk, Milan, Italy) and propagated twice in MRS (mMRS) modified by maltose and fresh yeast extract supplementation at 1 and 5 g/100 mL, respectively (final pH 5.60), before experimental use.

### 2.2. Technological Characterization of the Strains

Strains were screened for urease activity according to Mora et al. [13]. Starch hydrolysis was assessed on drop-inoculated starch agar plates (KNO_3_ 0.5 g/L, K_2_HPO_4_ 1 g/L, MgSO_4_.7H_2_O 0.2 g/L, CaCl_2_ 0.1 g/L, FeCl_3_ traces, potato starch 10 g/L, agar 15 g/L). Plates were incubated at 30 °C up to colonies’ visible growth. The amylolytic activity was determined by detecting clear zones around colonies after staining with Lugol’s solution. Phytase activity was assessed according to Anastasio et al. [14] in modified Chalmers medium. Exopolysaccharide (EPS) production, as well as sugars’ fermentation, were evaluated according to Blaiotta et al. [15]. β-glucosidase activity was assayed by using 4-nitrophenyl β-D glucopyranoside (Fluka, Milan, Italy) according to the method proposed by Fia et al. [16]. 

The fermentative abilities of the strains were assessed in terms of acids and gas development [17]. LAB loads, as well as volume increase, pH, and Total Titratable Acidity (TTA) were evaluated at time zero and after 24 h of incubation at 28 °C. In detail, LAB were counted by drop method [18] onto mMRS agar plates; pH values were determined by a lab pH-meter (XS, model pH 50); TTA was measured on 10 g of sample homogenized with 90 mL of distilled water and expressed as the amount (mL) of 0.1 mol/L NaOH required to reach the pH of 8.5.

### 2.3. Fermentation Abilities of Sourdough Starters on a Laboratory Scale

Three starter combinations were selected and used for further trials. MRS cultures in exponential phase of growth were mixed in a 2:1 (lactobacilli–pediococci) ratio; specifically, 80 mL of each *Lactiplantibacillus* (*L.*) *plantarum* strain (alternatively E73, E75 or C710) was mixed with 40 mL of the *Pediococcus* (*P.*) *lolii* strain B72. Bakery yeast (12 mL) was added as pure culture in Malt Extract (ME) broth (Oxoid) after incubation at 28 °C for 24 h. Cells were collected by centrifugation (10 min at 14,000 rpm), washed with sterile saline, and resuspended in an equal volume (132 mL) of sterile tap water. The cell suspensions were then mixed with 200 g of wheat flour (Barilla S.p.A., Parma, Italy) under aseptic conditions to get the dough (initial LAB loads around 8 Log CFU/g). After 16 h of incubation at 28 °C, the height (cm) reached by the dough in a graduated tube was recorded. Sourdough samples were subject to pH and TTA evaluation. LAB were counted by drop method onto differential MRS (dMRS) agar [19] plates supplemented with cycloheximide (0.1 g/L) at 30 °C for 48 h, while the number of yeasts was estimated on ME Agar supplemented with chloramphenicol (0.1 g/L) at 28 °C for 72 h. The set of trials was repeated by using the same amount of flour but with microbial population levels ten- or one hundred-fold lower. 

CO_2_ development was evaluated by placing the doughs into tightly sealed bottles whose lids were equipped with a manometer for the measurement of the gas development. The initial pressure inside the bottle at the start of fermentation was recorded. The value was subtracted from that obtained at the end of the incubation at 28 °C for 16 h.

### 2.4. Use of Sourdough Starters for Bread Production

Doughs were prepared by using the previously described microbial combinations with an additional control sample consisting of bakery yeast only. Experiments were carried out by using an initial LAB inoculum of about 7 Log CFU/g. Four hundred grams of wheat flour was mixed with 215 mL of sterile tap water by using an electronic mixer. The doughs were then incubated at 28 °C for 16 h. Samples were collected at time zero and after 16 h of fermentation for bacteria and yeast counts, as well as for pH and TTA measurements as detailed in Section 2.1. Bread and dough samples at time 0 as well as at the end of fermentation were even subject to volatile organic compounds (VOCs) analysis according to Aponte et al. [11]. For the analyses, a silica fiber, coated with 85 μm of Carboxen–polydimethylsiloxane (Carboxen/PDMS) was used (Supelco, Bellefonte, PA, USA).

VOCs analysis was performed using an Agilent Technologies (Santa Clara, CA, USA) 7890A gas chromatograph coupled to an Agilent Technologies 5975 mass spectrometer equipped with a 30 m × 0.25 mm ID, film thickness 0.25 µm capillary column (HP-INNOWAX, Agilent Technologies, Santa Clara, CA, USA). The gas carrier was helium (flow 1.5 mL/min) and SPME injections were splitless (straight glass line, 0.75 mm ID) at 240 °C for 20 min during which time thermal desorption of analytes from the fiber occurred. The oven parameters were as follows: the initial temperature was 40 °C held for 3 min, followed by an increase to 240 °C at a rate of 5 °C/min, then held for 10 min. Mass spectrometer operated in scan mode from m/z 33–300 (2 s/scan) at an ionization potential of 70 eV.

Identification of volatiles was achieved by comparing mass spectra with the Wiley library (Wiley7, NIST 05). VOCs were identified by matching the retention indices (RI) calculated according to Van den Dool and Kratz [20] and De Luca et al. [21] by the internal standard method. Results are expressed as mg/g of sample. Blank experiments were conducted in two different modalities: blank of the fiber and blank of the empty vial. These types of control were carried out after every 20 analyses. All the analyses were performed in triplicate.

### 2.5. Wet Granulation of Sourdough Starters

Granules containing each type of the bacterial/yeast combination were prepared by wet granulation according to Aponte et al. [9]. Eight hundred milliliters of MRS broth cultures of each *Lactiplantibacillus* strain were added to 400 mL of the *P. lolii* B72. Bakery yeast—120 mL of ME broth culture—was also added. Cells were collected by centrifugation, washed with sterile saline, and weighed. Alive cell counts were determined before the granulation process by drop counts onto dMRS and ME agar plates. 

For the preparation of the granules, pharmaceutical grade corn starch (CS), microcrystalline cellulose (MC), and lactose monohydrate (LM) at different weight ratios was used as diluents after sieving through a 50-mesh sieve (300 μm). When needed, aqueous polyvinylpyrrolidone (PVP) (2% *v*/*v*) was employed as binder. About 7 mL of aqueous PVP were mixed with 3.2 g of CS, 3.2 g of MC, and 3.8 g of LM and gradually added to cells collected by centrifugation and weighted (12.6 ± 0.13, 11.8 ± 0.42, 12.8 ± 0.28, and 9.3 ± 0.09 CFU/g for E75-B72, E73-B72, C710-B72, and yeast, respectively). Excipients were mixed in the mortar until a cohesive mass was formed. The obtained paste was passed through a 12-mesh sieve (1.7 mm) and dried in an oven at 30 °C for 12 h. After drying, all granules were screened through a 0.71, 1.0, and 1.4-mesh sieves and stored at 4 °C until use. To prevent any microbial contamination, filtered ultrapure water (Millex^®^ 0.22 μm sterile syringe filters, Millipore, Merck KGaA, Darmstadt, Germania) was employed throughout the experiment. All critical stages in the manufacture were carried out in a Grade A laminar flow hood [9].

Bacterial viability per gram was assessed according to Romano et al. [22]. Specifically, total cells were direct counted at the microscope by means of a Petroff-Hausser chamber, whereas alive cells were determined by plating onto dMRS agar plates. Living cells were also assessed by using the Live/dead BacLightTM Bacterial Viability Kit (Molecular Probes, Eugene, OR, USA) and subsequent observation under an epifluorescence microscope.

The fermentation ability of the granules was tested by mixing 1 g of each microbial combination with 10 g of flour. The test was performed by using fresh cells as well. Doughs were incubated at 28 °C for 16 h. Samples were taken at time 0 and after 16 h of fermentation for microbial counts and for the evaluation of organic acids and reducing sugars by HPLC analysis according to Blaiotta et al. [23]. The height in (cm) reached by each dough was also recorded.

### 2.6. Freeze-Drying of Mature Sourdoughs

Mature sourdoughs prepared by using the E75-B72 combination were freeze-dried after 16 h of fermentation at 28 °C. Doughs were frozen at −80 °C as they were, and after dilution 1/10 (*v*/*v*) with sterile Ringer solution. In both cases, doughs were spread in a thin layer on trays. The drying process was stopped only when the samples attained a constant humidity (3% w.b.). The freeze-dried sourdoughs (10% wt/wt) were used to make bread. Fresh cells of the same E75-B72 combination served as controls. pH, TTA, microbial loads, as well as the height reached by doughs after 16 h of fermentation at 28 °C, were assessed.

### 2.7. Sensory Analysis of Bread

Four bread samples were prepared with sourdoughs of the combination E75-B72 used as fresh (B) or wet granulated cells (D). Bakery yeast alone served as negative control (A), while a natural sourdough collected from an artisan bakery located in Naples (Campania Region, Italy) was used as a positive control (C). The four types of bread were subjected to sensory analysis performed by a descriptive panel composed of eighteen tasters (10 females and 8 males) recruited among students and all of whom were familiar with sensory analysis of foods. All respondents have consented to participation in the study. Eighteen descriptors were selected from those reported by Comendador et al. [24]. Sensory attributes used to describe bread quality could be categorized as follows: appearance and texture attributes referring to crust and crumb, odor and taste attributes referring to bread slices, and overall rating. The tasters rated the intensity of each attribute with a score on a 6-point hedonic scale (5 = extremely high; 0 = extremely low). Samples were coded and presented to assessors in random order.

### 2.8. Statistical Analysis

Significant differences among data were computed by using ANOVA and Tukey *t*-test (*p* < 0.05) (XLStat 2012.6.02 statistical pocket, Addinsoft Corp., Paris, France).

## 3. Results

### 3.1. Strains Selection

Forty-three strains were analyzed for biochemical features of technological interest. The production of lactic and acetic acids associated with sourdough fermentation may adversely affect LAB metabolic performances. The release of ammonia and carbon dioxide associated with urease activity seems to protect microorganisms against the harmful effects of acids [25]. The pH modulation capability of urease has been demonstrated in *Streptococcus thermophilus* [15]. In the present survey, only *P. lolii* strains expressed ureolytic activity (Appendix A: Genotypical (by Aponte et al., 2013), and phenotypical features of the 43 analyzed LAB strains). Zotta et al. [26] also failed to find lactobacilli able of hydrolyzing urea in sourdoughs used to produce ‘Cornetto di Matera’. Moreover, none of the tested strains expressed β-glucosidasic activity (data not shown). About 44% of the cultures were capable of degrading phytic acid. Most of the total phosphorus in cereals comes in the form of phytic acid (myo-inositol hexaphosphate): an anti-nutritional compound that chelates proteins, amino acids, and divalent cations such as Ca^2+^, Fe^2+^, Mg^2+^, Zn^2+^, preventing their uptake by the intestinal mucosa [26]. Consequently, phytates’ degradation by fermentation is certainly attractive. The 19% of the strains, all belonging to the *L. plantarum* group, hydrolyzed starch, and about 40% produced EPS in presence of this compound (Appendix A). On the other hand, all tested strains were able to produce EPS in presence of glucose, maltose, saccharose, and fructose. Regarding the use of pentoses as the sole carbon source, only one strain, *L. plantarum* E73, fermented arabinose, whereas six strains (B72, D71, D72, D74, E72, and F73) were able to use xylose. Fructose and fructose plus maltose were metabolized by all strains, while only two strains, both belonging to the *P. lolii* species (B72 and D73), were unable to metabolize maltose (Appendix A). Results partially matched those obtained by Bartkiene et al. [27] for LAB strains isolated from rye sourdoughs. According to the authors, only *P. acidilactici* could ferment xylose, but not maltose, while *L. plantarum* exhibited the opposite behavior. By combining strain typing at both phenotypical and molecular [11] levels, a sub-stratification can be noticed. In some cases, i.e., strain E73, patterns are unique at both levels. In other cases, strains isolated from different sourdoughs (coded with different letters) but characterized by the same DNA fingerprinting also share the same biochemical profile, e.g., *L. plantarum* strains F74 and D74 or F72, B71, and B75 (Appendix A). 

The strains were individually inoculated in 50 g wheat doughs under the same conditions to evaluate their effectiveness as starters for whole-wheat flour fermentation. Strains were classified as good (11), fair (13) or bad (19) fermenters based on the results (data not shown). TTA, pH, and counts on MRS for the best cultures are reported in Appendix A. Values recorded for pH and LAB loads did not significantly differ within strains, while TTA, and above all the height reached by the doughs, were quite variable. In detail, strains E73, C76, C710, E75, and F76 allowed reaching the highest heights. 

### 3.2. Fermentation Abilities of Selected Starters

Three lactobacilli were chosen from among the cultures designated as good fermenters based on a combination of biochemical characteristics and molecular strain typing. In detail, *P. lolii* B72 (xylose-fermenting with urease activity and unable to use maltose) was mixed with *L. plantarum* E75 (EPS producer by starch with phytasic activity), or *L. plantarum* C710 (highest leavening activity), or *L*. *plantarum* E73 (arabinose-fermenting) (Appendix A). In the first set of trials, the three selected combinations were evaluated for the ability to ferment wheat flour by using three levels of inocula: around 8 (H), 7 (M), and 6 (L) Log CFU/g. Initial and final levels (after 16 h) of LAB, as well as of lactobacilli and pediococci separately counted on dMRS medium, are reported in Table 1. pH, TTA, and the height reached by the dough after 16 h of fermentation at 28 °C were reported too. 

Despite the three different inocula, LAB populations always reached levels around 9 Log CFU/g after 16 h of fermentation and the yeast grew by nearly 2 Logs for all combinations. On the other hand, the height reached after 16 h of incubation proved to be quite variable. The highest height was recorded in the sourdough containing the combination E75-B72 which was 5.5 cm when using the highest inoculum (H). No significant difference in leavening was observed when using a 10-fold lower (M) bacterial inoculum for combination E75-B72 (*p* > 0.05) (Table 1). pH values were similar in all combinations at time zero, ranging from 5.68 to 6.10. After 16 h of fermentation, there was a considerable reduction in pH, with values ranging from 3.54 to 3.76. Lim et al. [28] found similar results: different LAB starters reduced pH by up to 3.5–3.9. TTA levels that reached an average of 12 mL of NaOH after fermentation likewise showed this variance (Table 1). Following the results, an intermediate level of inoculum (M) was chosen for future studies. The best performance of the E75-B72 combination was validated by measuring gas production after 16 h of fermentation (Appendix A).

### 3.3. VOCs Determination in Sourdoughs and Bread

The three microbial combinations (E73-B72, E75-B72, and C719-B72) were used to prepare bread. Level M was used as the initial inoculum. In all cases, starting from 7.5–7.6 Logs, the LAB population level reached about 8.4 Log CFU/g in the dough after 16 h of fermentation (Appendix A). Yeast counts were assessed before and after fermentation, by counting on ME medium. As expected, initial yeast loads were nearly identical (5 Log CFU/g) for all combinations including the control (yeast alone) and increased by around 2 Logs during fermentation. The combination E75-B72, which appeared to be the most promising, achieved the highest height (Appendix A).

Although the aroma is mainly produced upon baking, the generation of several precursors occurs during fermentation. Sourdough bread VOCs produced along with fermentation by biological and biochemical actions have been proved to intensely affect the sensory profile [29]. The four samples were evaluated using SPME-GC/MS at time zero, after 16 h of fermentation, and after baking to track the VOCs formation. A total of 21 VOCs were found, but only those detected at a concentration higher than 0.5 mg/g. are listed in Table 2. 

In doughs at time 0, ethyl acetate was the sole detected VOC in sample inoculated with combinations E73-B72 (1.23 ± 0.02 mg/g) and C710-B72 (0.07 ± 0.00 mg/g) (data not shown). According to the findings, 3-methyl-1-butanol was identified in all sourdoughs, however, 2-methyl-1-propanol was only found in sourdough and bread made with the E75-B72 and C710-B72 combinations. Both compounds have been linked to homofermentative fermentation [30]. The concentration of 2-methyl 1-propanol, 3-methyl 1-butanol, and ethyl acetate in sourdough bread is to be connected to their contents in the corresponding sourdough [30] and is responsible for alcoholic, fruity, malty, and acid odor, assuring a more intense aroma [4]. Ethyl acetate disappeared after baking (Table 2). However, esters’ disappearing due to evaporation during baking has been often reported [29,31]. On the other hand, during baking, the number of VOCs from different origins can increase. For example, the content of 2- and 3-methylbutanal may rise [29] as here reported (Table 2). This increase in bread is probably due to the free amino acids brought by sourdough into bread, which then contribute to the Maillard reaction during baking. Specifically, 2-methylpropanal, 2-methylbutanal, and 3-methylbutanal derive from yeast conversion of valine, isoleucine, and leucine, respectively [30].

Aldehydes, such as hexanal, could result from the degradative oxidation of unsaturated fatty acids. According to data, hexanal was present in all sourdough after 16 h of fermentation but just in two sourdough bread (Table 2). These findings are consistent with those reported in several studies [32,33,34]. At any rate, lipid oxidation compounds are frequently characterized as being off-flavors. Therefore, the high content of hexanal in fermented bread may result in a lower product acceptance [21].

It is well known that some volatiles found in bread can be formed by more than one route, which sometimes complicates the interpretation of the effects of processing factors: for example, the aldehydes 2-methylpropanal, 2- and 3-methylbutanal, can be formed via the Ehrlich pathway during the fermentation step or via Strecker degradation during baking [35]. Similar results were also obtained by Gobbetti et al. [36]: in sourdough made by combining lactobacilli and *Saccharomyces exiguous*, 2-methyl 1-propanal appeared after baking. Similarly, 2-methyl-1- propanal and 2- and 3-methyl-l-butanal were mostly found in LAB and *S. exiguous* sourdough bread [36]. 

Two important fermentation indexes, ethanol and acetoin, were not retrieved. 2,3-butanedione was detected only in two sourdough bread, but not in the corresponding mature dough. Actually, this compound can be formed either through glycolysis of pyruvic acid, asparagine conversion by yeast, or during baking due to Maillard reaction [35]. Overall, when compared to bakery yeast bread, sourdough bread produced under these conditions was characterized by more complex aroma profiles. According to earlier studies [30,31], a high concentration of 2,3-butanedione and 2-pentylfuran, both of which are characteristic of sourdough bread, suggests that the starter has a beneficial influence on the sensory profile of bread.

### 3.4. Wet Granulation Process

By wet granulation of the three bacterial combinations plus the yeast alone, four size fractions were obtained: <0.710, 0.710–1, 1–1.4, and >1.4 mm. When subject to wet granulation alone, bakery yeast counts dropped from 9.30 ± 0.34 Log CFU/g to around 7 Log CFU/g independently by the fraction size (data not shown). The population level of cultures before wet granulation was compared to the LAB and yeast counts in granules, split by size (Figure 1).

Before granulation, bacterial loads ranged from 9.9 to 10.6 Log CFU/mL for all combinations, while yeast counts were in the range 7.6–8.0 Logs. Even though differences were not significant t (*p* > 0.05), cells survival appeared to be directly correlated to granules size: the larger the granules, the higher the cell survival, likely due to the matrix’s ability to entrap microorganisms more safely (Figure 1).

For the quantification of cells not injured by wet granulation, the granules of the four sizes were accurately mixed. According to data, bacterial combinations significantly differed in their ability to withstand granulation (*p* < 0.05), with E75-B72 surviving the process better than the other two. The count of green viable bacterial cells by Live/dead BacLightTM Bacterial Viability Kit was constantly higher than the number of colonies obtained by counting on MRS medium (Figure 2). 

The three combinations were tested for their fermentation abilities as fresh cells and as granules. The amount of granules to use as starter was calculated in order to ensure nearly the same number of LAB and yeast usually used as wet cells (Table 3). Actually, the initial microbial loads for wet and granulated cells never coincided. After 16 h of fermentation, microbial population levels, as well as pH and TTA, did not show statistically significant differences (*p* > 0.05), except for yeast and pediococci in sourdoughs E75-B72 and E73-B72, respectively. Conversely, the height reached by the various mature sourdoughs was the most impressive result: it was constantly higher (*p* < 0.05) in doughs inoculated with granules rather than fresh cells (Table 3).

Reducing sugars and organic acids variations during sourdough fermentations were assessed by HPLC analyses at time zero and after 16 h of fermentation (Table 4).

Glucose levels ranged by 2–3 g/kg in all doughs at the onset of fermentation and as expected, quickly declined. The lowest values were recorded for the control and the starter C710-B72 (Table 4). The same trend could be noticed for fructose as well. Surprisingly, the content of maltose at the end of fermentation was higher than that recorded in doughs at the start of fermentation. Indeed, such a phenomenon has already been described [37]. Maltose content can increase during sourdough fermentation due to the hydrolytic activity of indigenous amylases on the starch fraction damaged during the milling process [38]. Gobbetti et al. [39] achieved similar results using *L. plantarum* strain as starter. 

Due to the utilization of fermentable carbohydrates and the growth in LAB population, lactic acid—nearly absent before fermentation—considerably increased in all combinations, except in the control dough. After 16 h of fermentation, the values were about 7–11 g/kg. Of course, this may explain the pH drop that has been observed (Table 3). Ethanol content was absent in all combinations at the beginning of fermentation, as expected, but its concentration after fermentation proved to be extremely variable within analyzed samples. In detail, the ethanol content was always significantly higher (*p* < 0.05) in sourdoughs made using wet granulated starters (Table 4). Since the CO_2_ development is directly related to the ethanol amount, HPLC results may explain why sourdoughs inoculated with wet granulated cells attain a higher height (Table 3).

### 3.5. Sensory Evaluation

The results of the sensory tests on the bread are reported in Figure 3. With reference to the appearance of the crust, the results of comparing bread from different trials revealed noticeable differences. Bread produced by using the traditional sourdough was almost identical to bread made with the starter, regardless of the formulation type. When compared to bread made with bakery yeast only, the three kinds of bread produced with sourdough proved to be characterized by a crispier and thicker crust, as well as a more intense color (*p* < 0.05). The bread made with bakery yeast also presented a more elastic crust. Instead, no difference was recorded with reference to the crumb: only bread made only with yeast had a gummier crumb (Figure 3). 

Regarding odor, the sourdough descriptor was perceivable only in bread made with natural sourdough. By evaluating taste rating data, substantial differences could be retrieved. Because no salt was added to the dough, all samples—without exceptions—were deemed unsatisfactory for the salt. The bread produced by adopting the starter as fresh cells was bitterer and more acidic (Figure 3). This trait is appreciated by Italian consumers since it is associated with traditional bread. According to the overall acceptance rating, bread produced by using the combination E75-B72 was the most appreciated by panelists, but only when the starter was added to the dough as fresh cells. The samples less appreciated were those produced by using the bakery yeast only (Figure 3).

### 3.6. Freeze-Drying Effect on the Viability of Sourdough Bacteria

Mature sourdoughs obtained by using the combination E75-B72 were subject to freeze-drying. LAB and Yeast population levels in sourdoughs were around 8.7 and 7.2 Log CFU/g, respectively. After drying, microbial counts declined by nearly one Log for LAB (around 7) and by about 2 Logs up to 5.3 for yeasts (data not shown). Powders were used at 10% as inoculum to lead fermentation. Bacterial and yeast loads were able to increase by roughly 2 logs (Figure 4). Noticeable differences (*p* < 0.05) emerged with reference to leavening: the height of sourdoughs made with the freeze-dried starter was 1.9 ± 0.21 cm, compared to 3.9 ± 0.34 for sourdoughs made with the same starter but employed as fresh cells.

## 4. Conclusions

The most common microorganisms applied to sourdough bread fermentation are LAB, and the selection of reliable starter cultures in sourdough bread-making is still a pivotal issue. The use of starters holds great prospects to improve the quality of cereal fermented foods. To achieve this, an accurate screening approach was adopted to select the best performing combination of strains. Wet granulation was used for the very first time in a food system and proved to be a reliable, fast, and low-cost method to stabilize starter cultures, even if some further work is needed to improve the taste of the bread produced. According to findings, the sourdough starter named E75-B72 has the potential to be marketed in the form of lightweight, and stable, wet granulated preparation. This combination of strains allows the convenient, direct manufacturing of baked goods with constant quality in 16 h while retaining the benefits of the biological fermentation process. With the aim of industrial purposes, the assessment of the cell viability retainment of the wet granulated sourdough during storage is certainly essential.

## Figures and Tables

**Figure 1 foods-11-01278-f001:**
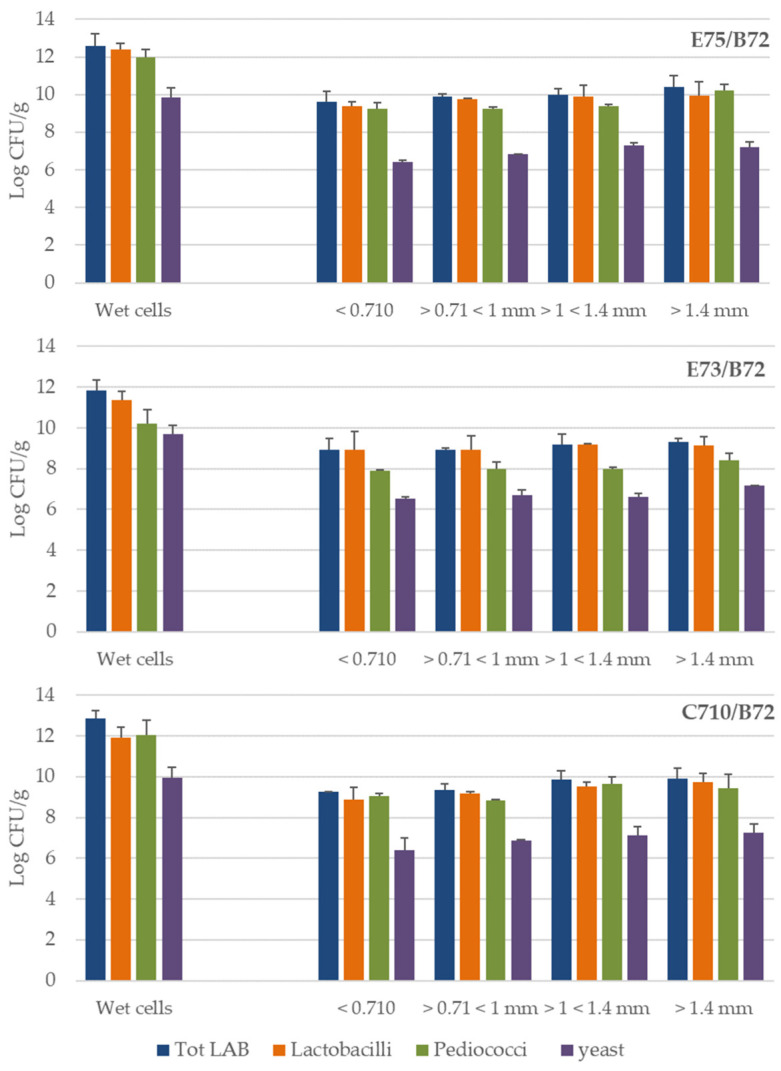
Survival of LAB and yeast after wet granulation compared with the initial population level (wet cells) for each granules’ size <0.710, 0.710–1, 1–1.4, and >1.4 mm. Analyses were performed in triplicate. Results are reported as mean values ± SD.

**Figure 2 foods-11-01278-f002:**
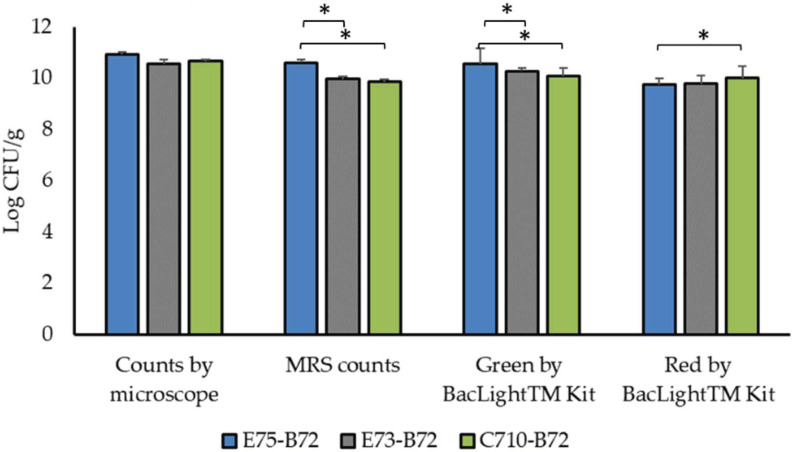
Microbial viability (Log CFU/g) after wet granulation. MRS counts were done in triplicate and counts at the microscope were repeated five times. Results are reported as mean values ± SD. Asterisks indicate significant differences among strains’ combinations (*p* < 0.05).

**Figure 3 foods-11-01278-f003:**
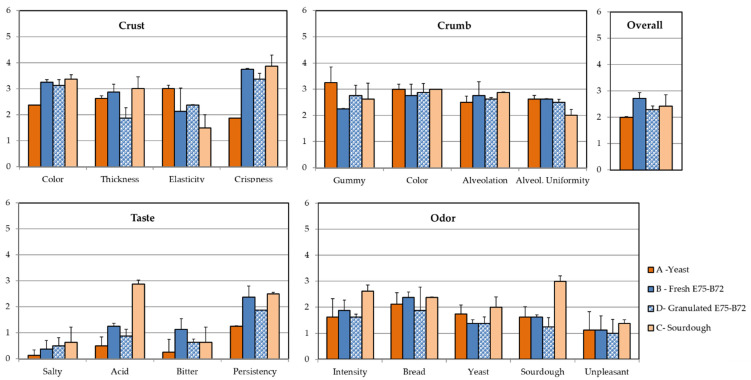
Sensorial descriptors mean scores and standard deviations of bread produced with natural sourdough, bakery yeast, or the starter E75-B72 as fresh or granulated cells.

**Figure 4 foods-11-01278-f004:**
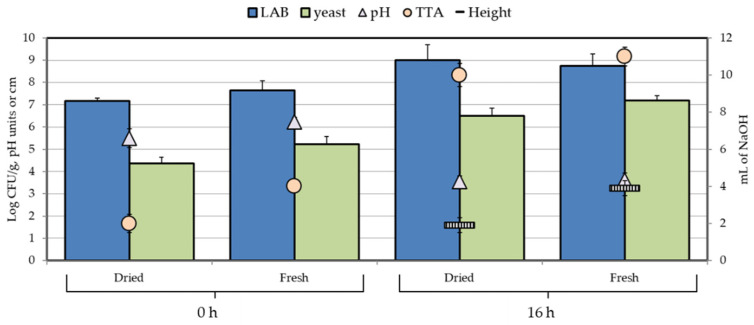
Microbial counts, pH, and TTA in sourdough produced with starter culture E75-B72 as fresh or freeze-dried cells at time 0 and after 16 h of fermentation at 28 °C. Height is reported as dashed bars. Analyses were performed in triplicate. Results are reported as mean values ± SD.

**Table 1 foods-11-01278-t001:** Microbial counts, pH, TTA, and height at time 0 and after 16 h of fermentation at 28 °C. LAB and yeast starter inoculated in wheat doughs at about 8 [H], 7 [M], and 6 [L] Log CFU/g. Analyses were performed in triplicate. Results are reported as mean values ± SD. Different superscript letters in a column indicate statistically significant differences (ANOVA: Tukey *t*-test. *p* < 0.05. XLStat).

	Trial	Time 0			After 16 h		
		E75-B72	E73-B72	C710-B72	E75-B72	E73-B72	C710-B72
LAB (CFU/g)	[H]	7.92 ± 0.00 ^a^	8.18 ± 0.42 ^a^	8.12 ± 0.02 ^a^	9.32 ± 0.40 ^a^	9.22 ± 0.09 ^a^	9.37 ± 0.11 ^a^
	[M]	7.28 ± 0.10 ^b^	7.56 ± 0.23 ^b^	7.12 ± 0.19 ^b^	9.06 ± 0.60 ^a^	9.12 ± 0.00 ^a^	9.22 ± 0.32 ^a^
	[L]	6.39 ± 0.22 ^c^	6.35 ± 0.14 ^c^	6.25 ± 0.44 ^c^	8.69 ± 0.70 ^a^	9.10 ± 0.55 ^a^	9.00 ± 0.05 ^a^
Lactobacilli (CFU/g)	[H]	7.82 ± 0.16 ^a^	8.03 ± 0.90 ^a^	7.88 ± 0.34 ^a^	9.07 ± 0.34 ^a^	9.15 ± 0.21 ^a^	9.12 ± 0.05 ^a^
	[M]	7.24 ± 0.00 ^b^	7.67 ± 0.02 ^a^	7.09 ± 0.01 ^b^	9.03 ± 0.25 ^a^	8.87 ± 0.33 ^a^	9.17 ± 0.03 ^a^
	[L]	5.31 ± 0.01 ^c^	6.30 ± 0.50 ^b^	6.80 ± 0.02 ^c^	8.38 ± 0.01 ^b^	8.90 ± 0.01 ^a^	8.80 ± 0.43 ^a^
Pediococci (CFU/g)	[H]	7.42 ± 0.30 ^a^	7.62 ± 0.01 ^a^	7.77 ± 0.60 ^a^	8.96 ± 0.00 ^a^	8.40 ± 0.00 ^a^	9.00 ± 0.21 ^a^
	[M]	7.22 ± 0.40 ^a^	6.22 ± 0.06 ^b^	6.62 ± 0.15 ^b^	7.92 ± 0.40 ^b^	8.76 ± 0.40 ^a^	8.22 ± 0.34 ^b^
	[L]	5.90 ± 0.21 ^b^	5.80 ± 0.16 ^c^	5.70 ± 0.60 ^c^	8.31 ± 0.30 ^b^	8.60 ± 0.50 ^a^	8.40 ± 0.42 ^b^
Yeast (CFU/g)	[H]	6.00 ± 0.14 ^a^	5.80 ± 0.16 ^a^	6.00 ± 0.24 ^a^	7.80 ± 0.33 ^a^	7.00 ± 0.55 ^a^	7.50 ± 0.01 ^a^
	[M]	5.00 ± 0.02 ^b^	4.70 ± 0.23 ^b^	5.00 ± 0.54 ^a^	7.10 ± 0.60 ^b^	7.00 ± 0.32 ^a^	7.10 ± 0.90 ^a^
	[L]	4.30 ± 0.11 ^c^	3.20 ± 0.22 ^c^	3.60 ± 0.05 ^b^	7.00 ± 0.01 ^b^	6.20 ± 0.60 ^b^	7.20 ± 0.06 ^a^
TTA (mL NaOH)	[H]	5.00 ± 0.10 ^a^	5.00 ± 0.23 ^a^	4.00 ± 0.59 ^a^	12.00 ± 0.32 ^a^	15.00 ± 0.15 ^a^	14.00 ± 0.32 ^a^
	[M]	4.00 ± 0.22 ^b^	3.00 ± 0.14 ^b^	3.00 ± 0.44 ^a,b^	11.00 ± 0.37 ^b^	12.00 ± 0.55 ^b^	10.00 ± 0.05 ^c^
	[L]	3.00 ± 0.16 ^c^	2.00 ± 0.34 ^c^	2.50 ± 0.34 ^b^	11.80 ± 0.34 ^a^	11.30 ± 0.21 ^c^	11.00 ± 0.05 ^b^
pH (units)	[H]	5.68 ± 0.30 ^a^	5.90 ± 0.02 ^a^	5.90 ± 0.61 ^a^	3.55 ± 0.25 ^a^	3.54 ± 0.33 ^a^	3.60 ± 0.03 ^a^
	[M]	5.88 ± 0.02 ^a^	5.97 ± 0.03 ^a^	6.04 ± 0.54 ^a^	3.74 ± 0.60 ^a^	3.74 ± 0.32 ^a^	3.76 ± 0.20 ^a^
	[L]	5.98 ± 0.11 ^a^	6.09 ± 0.22 ^a^	6.10 ± 0.05 ^a^	3.65 ± 0.01 ^a^	3.62 ± 0.23 ^a^	3.50 ± 0.06 ^a^
Height (cm)	[H]				5.50 ± 0.39 ^a^	4.20 ± 0.34 ^a^	4.70 ± 0.40 ^a^
	[M]				5.20 ± 0.42 ^a^	3.60 ± 0.12 ^b^	3.10 ± 0.35 ^b^
	[L]				4.00 ± 0.24 ^c^	2.60 ± 0.26 ^c^	2.30 ± 0.21 ^c^

**Table 2 foods-11-01278-t002:** VOCs (mg/g) detected in doughs at time zero, sourdough after 16 h of fermentation, and bread samples. Only VOCs detected at a concentration higher than 0.5 mg/g are reported as mean values ± SD. Different superscript letters in a row indicate statistically significant differences (ANOVA: Tukey *t*-test. *p* < 0.05. XLStat).

	RI	Analytes	Odor	Dough after 16 h				Bread			
				C	E73-B72	E75-B72	C710-B72	C	E73-B72	E75-B72	C710-B72
Esters	905	Ethyl acetate	Fruity sweet	23.70 ± 1.76 ^d^	35.91 ± 0.86 ^c^	54.32 ± 1.02 ^a^	42.76±2.00 ^b^	nd ^a^	nd ^a^	nd ^a^	nd ^a^
Alcohols	1196	Ethanol	Alcoholic	6.81 ± 0.03 ^a^	nd ^b^	nd ^b^	nd ^b^	nd ^a^	nd ^a^	nd ^a^	nd ^a^
	1253	3-Methyl-1-butanol Isoamyl alcohol	Banana, fruity, almonds	38.75 ± 1.87 ^d^	78.52 ± 1.90 ^c^	127.30 ± 2.30 ^a^	113.20 ± 2.50 ^b^	27.18 ± 1.54 ^d^	65.64 ± 2.09 ^c^	100.9 ± 2.8 ^a^	89.41 ± 3.05 ^b^
	1097	2-Methyl-1-propanol Isobutyl alcohol	Ethereal	nd ^c^	nd ^c^	5.60 ± 0.02 ^b^	7.30 ± 0.14 ^a^	nd ^c^	nd ^c^	3.74 ± 0.00 ^b^	5.96 ± 0.12 ^a^
Ketones	1032	2,3-butanedione Diacetyl	Sweet, butter	nd ^a^	nd ^a^	nd ^a^	nd ^a^	nd ^c^	nd ^c^	7.78 ± 0.12 ^b^	9.23 ± 0.76 ^a^
Heterocyclic	1271	2 Pentylfuran	Green bean, raw nuts mushroom	nd ^a^	nd ^a^	nd ^a^	nd ^a^	nd ^c^	3.24 ± 0.01 ^a^	1.52 ± 0.03 ^b^	3.18 ± 0.03 ^a^
Aldehydes	1138	Hexanal	Fresh, green	nd ^c^	17.66 ± 0.08 ^a^	12.23 ± 1.54 ^b^	12.08 ± 1.00 ^b^	1.46 ± 0.31 ^c^	14.43 ± 0.00 ^a^	nd ^d^	7.48 ± 0.76 ^b^
Branched aldehydes	853	2 Methylpropanal Isobutyraldehyde	Malty	nd ^a^	nd ^a^	nd ^a^	nd ^a^	6.20 ± 0.03 ^a^	nd ^b^	nd ^b^	nd ^b^
	976	3-Methylbutanal Isovaleraldehyde	Malty, roasty cucumber-like	nd ^d^	3.23 ± 0.12 ^c^	4.99 ± 0.09 ^a^	3.78 ± 0.01 ^b^	12.71 ± 0.61 ^b^	15.80 ± 1.04 ^a^	16.14 ± 1.04 ^a^	14.32 ± 1.09 ^a^
	970	2-Methylbutanal 2-Methylbutyraldehyde	Nut, fruity	nd ^a^	nd ^a^	nd ^a^	nd ^a^	9.89 ± 0.73 ^a^	8.24 ± 0.87 ^a^	6.51 ± 0.48 ^b^	5.39 ± 0.65 ^c^
Acids	1515	Acetic acid	Sour, acid, pungent	12.64 ± 0.43 ^d^	23.51 ± 0.78 ^b^	32.67 ± 0.99 ^a^	15.44 ± 1.02 ^c^	11.60 ± 0.03 ^d^	20.08 ± 1.00 ^a^	31.01 ± 1.33 ^b^	13.22 ± 0.89 ^c^

**Table 3 foods-11-01278-t003:** Microbial counts (Log CFU/g), pH (units), TTA (mL of NaOH), and height (cm) of fresh (F) and granulated cells (G) at time zero and after 16 h of fermentation. Analyses were performed in triplicate. Results are reported as mean values ± SD. Different superscript letters in a row indicate statistically significant differences between F and G (*p* < 0.05).

Trial	Parameter	Time 0		Time 16	
		G	F	G	F
E75-B72	LAB	8.40 ± 0.10 ^a^	7.60 ± 0.23 ^b^	9.12 ± 0.19 ^a^	9.70 ± 0.60 ^a^
	Lactobacilli	8.30 ± 0.42 ^a^	7.40 ± 0.54 ^b^	8.80 ± 0.44 ^a^	9.60 ± 0.70 ^a^
	Pediococci	7.90 ± 0.16 ^a^	7.00 ± 0.90 ^a^	8.70 ± 0.34 ^a^	9.09 ± 0.34 ^a^
	Yeast	5.60 ± 0.00 ^a^	4.90 ± 0.02 ^b^	7.20 ± 0.01 ^a^	6.80 ± 0.25 ^b^
	pH	5.11 ± 0.03 ^b^	6.07 ± 0.54 ^a^	3.72 ± 0.60 ^a^	3.66 ± 0.32 ^a^
	TTA	3.01 ± 0.86 ^a^	2.02 ± 0.16 ^b^	15.10 ± 0.91 ^a^	16.08 ± 0.80 ^a^
	Height			3.90 ± 0.21 ^a^	2.70 ± 0.18 ^b^
E73-B72	LAB	7.35 ± 0.02 ^b^	7.66 ± 0.01 ^a^	9.80 ± 0.01 ^a^	9.00 ± 0.43 ^a^
	Lactobacilli	7.33 ± 0.60 ^a^	7.45 ± 0.80 ^a^	9.40 ± 0.70 ^a^	8.87 ± 0.61 ^a^
	Pediococci	5.92 ± 0.15 ^b^	7.26 ± 0.40 ^a^	9.50 ± 0.40 ^a^	8.39 ± 0.34 ^b^
	Yeast	5.22 ± 0.60 ^a^	4.41 ± 0.30 ^b^	7.30 ± 0.50 ^a^	6.50 ± 0.42 ^a^
	pH	5.88 ± 0.01 ^a^	5.88 ± 0.50 ^a^	3.70 ± 0.30 ^a^	3.74 ± 0.30 ^a^
	TTA	2.03 ± 0.96 ^a^	3.04 ± 0.94 ^a^	15.01 ± 0.82 ^a^	15.03 ± 0.72 ^a^
	Height			3.60 ± 0.34 ^a^	1.50 ± 0.11 ^b^
C710-B72	LAB	8.61 ± 0.23 ^a^	7.26 ± 0.19 ^b^	9.80 ± 0.60 ^a^	9.50 ± 0.15 ^a^
	Lactobacilli	7.57 ± 0.64 ^a^	7.09 ± 0.44 ^a^	9.60 ± 0.70 ^a^	9.40 ± 0.55 ^a^
	Pediococci	7.60 ± 0.90 ^a^	6.76 ± 0.54 ^a^	9.30 ± 0.64 ^a^	9.03 ± 0.21 ^a^
	Yeast	5.70 ± 0.02 ^a^	4.90 ± 0.01 ^b^	7.30 ± 0.25 ^a^	6.90 ± 0.33 ^a^
	pH	4.74 ± 0.21 ^b^	6.16 ± 0.11 ^a^	3.60 ± 0.40 ^a^	3.66 ± 0.03 ^a^
	TTA	3.03 ± 0.94 ^a^	2.00 ± 1.64 ^b^	13.03 ± 0.96 ^a^	12.10 ± 1.02 ^a^
	Height			3.70 ± 0.16 ^a^	2.20 ± 0.26 ^b^

**Table 4 foods-11-01278-t004:** Reducing sugars and organic acids (g/kg) by HPLC during sourdoughs fermentation at time 0 and after 16 h for fresh (F) and granulated (G) cells. Control is made with fresh bakery yeast. Analyses were performed in triplicate. Results are reported as mean values ± SD. Different superscript letters in a column indicate statistically significant differences (*p* < 0.05).

Trial		Glucose	Maltose	Fructose	Lactic acid	Ethanol
E75-B72	0	2.19 ± 0.16 ^a^	9.30 ± 0.90 ^c^	4.08 ± 0.34 ^a^	0.00 ± 0.00 ^c^	0.00 ± 0.00 ^c^
	F	0.82 ± 0.00 ^c^	16.87 ± 0.52 ^a^	1.41 ± 0.01 ^c^	7.46 ± 1.25 ^a^	1.49 ± 0.33 ^b^
	G	1.35 ± 0.01 ^b^	15.23 ± 0.50 ^b^	2.58 ± 0.02 ^b^	7.74 ± 0.91 ^a^	6.08 ± 0.41 ^a^
E73-B72	0	2.12 ± 0.22 ^a^	9.79 ± 0.84 ^c^	4.28 ± 0.44 ^a^	0.00 ± 0.00 ^b^	0.00 ± 0.00 ^c^
	F	1.52 ± 0.30 ^b^	19.55 ± 0.95 ^a^	2.41 ± 0.36 ^b^	7.48 ± 1.00 ^a^	1.20 ± 0.70 ^b^
	G	0.51 ± 0.10 ^c^	15.52 ± 0.43 ^b^	1.75 ± 0.34 ^c^	8.47 ± 0.60 ^a^	3.41 ± 0.67 ^a^
C710-B72	0	2.42 ± 0.10 ^a^	9.86 ± 0.23 ^b^	3.75 ± 0.19 ^b^	0.87 ± 0.60 ^c^	0.00 ± 0.00 ^c^
	F	0.00 ± 0.00 ^c^	16.11 ± 0.86 ^a^	1.27 ± 0.15 ^a^	9.18 ± 0.84 ^b^	3.93 ± 0.60 ^b^
	G	0.37 ± 0.21 ^b^	16.14 ± 0.96 ^a^	1.94 ± 0.60 ^a^	11.41 ± 0.63 ^a^	9.16 ± 0.50 ^a^
Control	0	2.87 ± 0.22 ^a^	11.79 ± 1.14 ^b^	5.42 ± 0.44 ^a^	0.00 ± 0.00 ^b^	0.00 ± 0.00 ^b^
	F	0.00 ± 0.00 ^b^	15.13 ± 0.62 ^a^	1.49 ± 0.21 ^b^	1.12 ± 0.55 ^a^	4.47 ± 0.33 ^a^

## Data Availability

Data is contained within the article or Appendix A.

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
