# Peer review of "Development of a Wet-Granulated Sourdough Multiple Starter for Direct Use"

_foods, 2022, doi:10.3390/foods11091278_

Round 1
Reviewer 1 Report
This study is interesting because the selection of starter culture in sourdough bread-making is a pivotal issue that holds great prospects to improve the quality of fermented foods. The study managed to screen and select the best performing combination of strains and used a wet granulation technique to stabilize the starter culture. In this study, several LAB strains isolated from mature sourdoughs were evaluated for features of technological interest and tested for the fermentation ability in model systems. The best combinations (E75-B72) were subject to several protocols in order to point out the best way to keep microbial viability upon drying. It was also proven that the sourdough breads produced under this study conditions were characterized by a more complex aroma profiles as compared to bakery yeast bread.
In order to improve the quality of this manuscript, the following must be corrected:
LINE 268: The authors indicated that a considerable drop in pH could be noticed after 16 h of fermentation, with values in the 268 range 3.54-3.76. However, this data is not shown anywhere in the manuscript.
There is no figure 5 and 6 in the manuscript (Typo mistake)
Time 0 and 16h data is not highlighted or shown in figure 7.
Section 5: Recommendations should be included under this section
English and grammar mistakes must be corrected. For example, lines: 11, 17,20,23,42,58,78,115,193,215-216,234,247,410, 438
Author Response
Reviewer 1
This study is interesting because the selection of starter culture in sourdough bread-making is a pivotal issue that holds great prospects to improve the quality of fermented foods. The study managed to screen and select the best performing combination of strains and used a wet granulation technique to stabilize the starter culture. In this study, several LAB strains isolated from mature sourdoughs were evaluated for features of technological interest and tested for the fermentation ability in model systems. The best combinations (E75-B72) were subject to several protocols in order to point out the best way to keep microbial viability upon drying. It was also proven that the sourdough breads produced under this study conditions were characterized by a more complex aroma profiles as compared to bakery yeast bread.
In order to improve the quality of this manuscript, the following must be corrected:
Dear reviewer, may we thank you warmly for having found merit in our work. Please find here below the corrections carried out to meet your requests.
LINE 268: The authors indicated that a considerable drop in pH could be noticed after 16 h of fermentation, with values in the range 3.54-3.76. However, this data is not shown anywhere in the manuscript.
Apologize. This is correct: in figure 1 pH values at the end of fermentation are unaccountably missing. At any rate, as per the editor’s suggestion, this figure has been changed to a table with pH values fully reported.
There is no figure 5 and 6 in the manuscript (Typo mistake)
Thank you. The figures and tables were all checked.
Time 0 and 16 h data is not highlighted or shown in figure 7.
Yes. Figure 7, now Figure 4 was corrected.
Section 5: Recommendations should be included under this section
Let’s hope to have understood what the reviewer meant with recommendations. At any rate, conclusions were improved and a sentence about potential applications of the findings was added (Lines 468-473).
English and grammar mistakes must be corrected. For example, lines: 11, 17,20,23,42,58,78,115,193,215-216,234,247,410, 438
All reported sentences were amended or entirely rephrased. Details are provided here below.
Line 11: The sentence has been entirely rewritten (lines 11-12)
Line 17: The sentence has been entirely rewritten (lines 15-17)
Line 20: The sentence has been entirely rewritten (lines 17-19)
Line23: The sentence has been entirely rewritten (lines 23-24)
Line 42: The sentence has been entirely rewritten (lines 41-42)
Line 58: The sentence has been entirely rewritten (lines 75-80)
Line 78: The sentence has been entirely rewritten (lines 101-102)
Line 115: The word ‘load’ has been changed with ‘inoculum’ (line 138)
Line 193: The entire paragraph was rewritten (Lines 217-218)
Lines 215-216: The sentence has been entirely rewritten (lines 238-239)
Line 234: The sentence has been entirely rewritten (lines 257-259)
Line 247: The sentence has been entirely rewritten (lines 268-271)
Line 410: The sentence has been entirely rewritten (lines 420-424)
Line 438: The sentence has been entirely rewritten (lines 450-451)

Reviewer 2 Report
This statement I advise to re-arrange: "The search of a sourdough starter stable, for direct use and able to carry out one-step fermentation is still to be considered a crucial issue".
"a more complex aroma profile" doesn't mean that better sensorial quality will be achieved.
This sentence doesn't fit the rest of the abstract "Wet granulation was used for the first time for a food starter stabilization". The structure of the abstract should be the aim, methods, results and conclusion.
what alkanes were used for retention indices?
"2.4 Bread production" is not the best title for this subchapter, I cannot see the bread was produced. Also in line 277, it is not correct to write "bread production"
what type of SPME fiber was used in the experiment. Any references for this method? No chemical standards or internal standard was used?
Who was involved in sensory analysis?
Line 291: cooking? shouldn't it be "baking"?
How you can write VOCs ng/g if no chemicals were used?
Table 1 there are significant errors in the names of chemical groups. It is not possible that only 7 compounds were identified in bread.
No statistical method is used. In case of this, the results cannot be compared.
Ref 13, 15, 27 and 32 and 35 are quite not up-to-date references.
Author Response
Reviewer 2
This statement I advise to re-arrange: "The search of a sourdough starter stable, for direct use and able to carry out one-step fermentation is still to be considered a crucial issue".
Such sentence was changed in the Abstract (Lines 11-12) as well as in the Introduction (Lines 77-80).
"a more complex aroma profile" doesn't mean that better sensorial quality will be achieved.
True. The sentence was modified in order to clarify. Please see lines 354-357-354.
This sentence doesn't fit the rest of the abstract "Wet granulation was used for the first time for a food starter stabilization". The structure of the abstract should be the aim, methods, results and conclusion.
The abstract was fully revised. The order of info is the conventional one, but conclusions still retain much more room than methods.
what alkanes were used for retention indices?
A mixture of aliphatic hydrocarbons (C8–C24) was used for retention indices.
"2.4 Bread production" is not the best title for this subchapter, I cannot see the bread was produced. Also in line 277, it is not correct to write "bread production"
We absolutely agree. Both titles have been changed.
what type of SPME fiber was used in the experiment.
For the analyses, a silica fibre, coated with 85 µm of Carboxen–polydimethylsiloxane (Carboxen/PDMS) was used. Info was added at lines 144-145
Any references for this method? No chemical standards or internal standard was used?
The method of Aponte et al. (2013) was used with 4-methyl-2-pentanol as internal standard (100 mg/L standard solution). The reference was added at line 143.
Who was involved in sensory analysis?
Details were added at line 208.
Line 291: cooking? shouldn't it be "baking"?
Sure… Corrected at line 310.
How you can write VOCs ng/g if no chemicals were used?
Internal standards were used. The text has been correct (line 157)
Table 1 there are significant errors in the names of chemical groups.
Table 1 (now table 2) has been checked and re-formatted.
It is not possible that only 7 compounds were identified in bread.
First, we must apologize. During reviewing we realized that there was a mistake. VOCs concentrations - mg/g – were wrongly reported as ng/g in the table caption. Apart from this, we decided to report only VOCs whose concentration was higher than 0.5 mg/g. This info has been reported in the Table 2 caption. Anyhow, if deemed appropriate, we can include all detected VOCs and not uniquely the most representative.
No statistical method is used. In case of this, the results cannot be compared.
All data were subject to ANOVA, so we are not quite sure to understand what the reviewer meant and, above all, we did not get if he is still talking about VOCs.
Ref 13, 15, 27 and 32 and 35 are quite not up-to-date references.
Collins &, Lyne (1989) and Van den Dool and Kratz (1963) are reported as methodological references and cannot be avoided. The papers by Damiani et al (1996), Gobbetti et al. (1994 & 1995) are certainly quite outdated, but unfortunately, we could not find more recent articles to comment on our results.
Reviewer 3 Report
The authors focused on the development of a wet-granulated sourdough multiple starter for direct use. Even though the analytical part is good, there is a problem, regarding the novelty of the work. Many research efforts have been conducted regarding the application of various LAB in sourdough bread production, as well as the application of powdered sourdoughs with freeze drying or spray drying. The authors are advised to present wet granulation better and in higher content in the text.
Even though they write in the Conclusion Section that : “Wet granulation was used for the very first time in a food system and proved to be a reliable, fast and low-cost system to stabilize the starter culture even if some further efforts are still needed to improve the taste of the produced bread”, surprisingly, this method is not referred nowhere in the Introduction Section, even for another food system and of course is not explained.
The Introduction Section needs reconstruction. Is not well organized and the authors did not manage to present the originality. In addition, the citations need enrichment in this Section. Especially the last paragraph of this Section is not detailed, neither exhibit the scope of this work in a simply and highlighted way. In addition, I do not know why reference 5 was added.
There are important and key VOC that did not determined. On the contrary, only very few VOC were determined, which cannot offer any important outcome by this and the respective discussion is not scientifically appropriate.
Author Response
Reviewer 3
The authors focused on the development of a wet-granulated sourdough multiple starter for direct use. Even though the analytical part is good, there is a problem, regarding the novelty of the work. Many research efforts have been conducted regarding the application of various LAB in sourdough bread production, as well as the application of powdered sourdoughs with freeze drying or spray drying. The authors are advised to present wet granulation better and in higher content in the text.
Even though they write in the Conclusion Section that : “Wet granulation was used for the very first time in a food system and proved to be a reliable, fast and low-cost system to stabilize the starter culture even if some further efforts are still needed to improve the taste of the produced bread”, surprisingly, this method is not referred nowhere in the Introduction Section, even for another food system and of course is not explained.
The Introduction Section needs reconstruction. Is not well organized and the authors did not manage to present the originality. In addition, the citations need enrichment in this Section. Especially the last paragraph of this Section is not detailed, neither exhibit the scope of this work in a simply and highlighted way. In addition, I do not know why reference 5 was added.
We fully agree with reviewer. In such light, Introduction has been extensively improved. Wet granulation and its applications have been described at lines 56-72. Additionally, details were added in the last part to better focus the scope of the work. With reference to reference 5, now 11, we must apologize. Actually, we should have better pointed out even at this stage that strains were isolated and genetically characterized during a previous survey. This info was added at lines 81-82. All can we say in our defence is that this aspect was properly highlighted in the section Material and Methods.
There are important and key VOC that did not determined. On the contrary, only very few VOC were determined, which cannot offer any important outcome by this and the respective discussion is not scientifically appropriate.
Also, in this case, we feel we must apologize. During reviewing we realized that there was a mistake. VOCs concentrations - mg/g – were wrongly reported as ng/g in the table caption. Apart from this, we decided to report only VOCs whose concentration was higher than 0.5 mg/g. This info has been reported in the Table 2 caption. Anyhow, if deemed appropriate, we can include all detected VOCs and not uniquely the most representative.
Round 2
Reviewer 1 Report
All my concerns have been addressed fully. Thank you
Author Response
Thank yoy
Reviewer 2 Report
The Authors indeed improved the text but some mistakes are still noticed.
Table 2-please once more carefully check the names of chemical classes e.g. "chetones" should be "ketones" etc. Moreover, there are no statistical findings.
Line 112. So you cannot write " Table 2 lists all identified VOCs.", because this is not true.
In subchapter 3.3 you should write how many in total VOCs did you found and then how many you selected.
Line 328: "bread... into bread" check to please this sentence, it doesn't sound good.
What is the difference between odor and aroma in the sensory study? I know flavor as a combination of taste and aroma/odor. This is quite surprising. Therefore, what means "unpleasant"? Figure 3 is very chaotic, "crust", "crumb" but the aroma and taste were for whole bread or also "crust" and "crumb"? Also, not always the standard deviations are presented.
Table 4 doesn't contain statistics.
Author Response
Reviewer 2
The Authors indeed improved the text but some mistakes are still noticed.
That is absolutely correct. We went over the manuscript again and again, finding countless errors. In fuchsia, all changes to the English language are highlighted.
Table 2-please once more carefully check the names of chemical classes e.g. "chetones" should be "ketones" etc. Moreover, there are no statistical findings.
Several mistakes were found, corrected, and marked in yellow. All results was subjected to ANOVA, but data were released neither on tables, nor figures. When ANOVA revealed significant differences (p 0.05) between samples, these were provided in the text.
Line 112. So you cannot write " Table 2 lists all identified VOCs.", because this is not true.
True. The sentence has been rewritten. Please refer to lines 304-305.
In subchapter 3.3 you should write how many in total VOCs did you found and then how many you selected.
This info was also added at lines 304-305.
Line 328: "bread... into bread" check to please this sentence, it doesn't sound good.
Correct. Lines 331-332
What is the difference between odor and aroma in the sensory study? I know flavor as a combination of taste and aroma/odor. This is quite surprising. Therefore, what means "unpleasant"? Figure 3 is very chaotic, "crust", "crumb" but the aroma and taste were for whole bread or also "crust" and "crumb"? Also, not always the standard deviations are presented.
Sorry. Obviously, it was a matter of odour and flavour. Despite this, after reanalyzing the data, we observed that there were few differences in odour and flavour judgment. Furthermore, the same descriptors were considered, thus generating even more confusion. As a result, we chose to eliminate the flavor descriptors from the analysis; nevertheless, this did not result in any significant changes in the text flux because this aspect of the sensory analysis did not produce any useful results.
Table 4 doesn't contain statistics.
Please refer to the prior response.

Reviewer 3 Report
No further comments.
Author Response
Thank you